# Polyhistor: Parameter-Efficient Multi-Task Adaptation for Dense Vision Tasks

**Yen-Cheng Liu**
Georgia Tech
ycliu@gatech.edu

**Chih-Yao Ma**
Meta
cyma@meta.com

**Junjiao Tian**
Georgia Tech
jtian73@gatech.edu

**Zijian He**
Meta
zijian@meta.com

**Zsolt Kira**
Georgia Tech
zkira@gatech.edu

## Abstract

Adapting large-scale pretrained models to various downstream tasks via fine-tuning is a standard method in machine learning. Recently, parameter-efficient fine-tuning methods show promise in adapting a pretrained model to different tasks while training only a few parameters. Despite their success, most existing methods are proposed in Natural Language Processing tasks with language Transformers, and adaptation to Computer Vision tasks with Vision Transformers remains under-explored, especially for dense vision tasks. Further, in multi-task settings, individually fine-tuning and storing separate models for different tasks is inefficient. In this work, we provide an extensive multi-task parameter-efficient benchmark and examine existing parameter-efficient fine-tuning NLP methods for vision tasks. Our results on four different dense vision tasks showed that existing methods cannot be efficiently integrated due to the hierarchical nature of the Hierarchical Vision Transformers. To overcome this issue, we propose *Polyhistor* and *Polyhistor-Lite*, consisting of *Decomposed HyperNetworks* and *Layer-wise Scaling Kernels*, to share information across different tasks with a few trainable parameters. This leads to favorable performance improvements against existing parameter-efficient methods while using fewer trainable parameters. Specifically, *Polyhistor* achieves competitive accuracy compared to the state-of-the-art while only use $\sim 10\%$ of their trainable parameters. Furthermore, our methods show larger performance gains when large networks and more pretraining data are used.

## 1 Introduction

Foundation models trained with large-scale datasets have shown the success of adapting to a variety of downstream NLP and vision tasks [1]. As the state-of-the-art foundation models grow to billion or even trillion parameter models [2, 3, 4, 5, 6], individually fine-tuning all parameters of the model wastes significant computational resources. Further, for multi-task models, both fine-tuning and storing separate models for multiple tasks become infeasible on devices with low computation resources.

To alleviate this issue, several works [7, 8, 9] have proposed *parameter-efficient* fine-tuning methods to derive a better trade-off between trainable parameters and accuracy on downstream tasks. By only training a small amount of parameters, these existing methods can substantially narrow the accuracy gap compared to the baseline that fine-tunes all parameters. However, these existing approaches mainly focus on NLP tasks [10, 11, 12] or single-task adaptation on image classification [9], and

---

**Polyhistor**: *someone gifted or learned in multiple disciplines.*

36th Conference on Neural Information Processing Systems (NeurIPS 2022).

their applicability to more complicated vision tasks is unclear. On the other hand, the single-task adaptation methods [7, 10, 11, 12, 8] still need to learn and store task-wise parameters, and the number of trainable parameters increase with respect to the number of tasks.

Therefore, in this paper, we first conduct a thorough study on **how the existing successful parameter-efficient methods on NLP tasks perform on vision tasks**, particularly on more challenging dense vision tasks (*e.g.,* semantic segmentation, normals estimation). Second, based on our findings, we then **design a novel parameter-efficient method for adaptation to multiple dense vision tasks**. Our method leverage shared modules across tasks and encourage the model to use shared information in a more parameter-efficient manner.

To start with, we first evaluate the existing parameter-efficient methods in NLP on dense vision problems. We chose to apply these methods to hierarchical vision transformers (HVTs) considering their state-of-the-art results on many per-pixel vision tasks [13, 14]. Through our extensive studies, we find **two** limitations in these works. First, adapter-based methods [11, 15], which have shown strong performance on NLP parameter-efficient adaptation benchmarks, cannot be efficiently integrated with HVTs. This is because the parameter usage of adapters in later transformer blocks grows quadratically with respect to the layer scale (see Fig. 1c). Second, the state-of-the-art multi-task parameter-efficient method [16] applies a hyper-network to produce the weights of adapters and shares information across different NLP tasks, while we find it inherently requires a large number of trainable parameters in the hyper-network (see Sec. 4.1 for further discussion).

To address the above limitations, we propose *Polyhistor-Lite*, which consist of two main components, *Decomposed Lite-HyperNetworks* and *Layer-wise Scaling Kernels*. These two methods reduce the trainable parameters in two aspects respectively, including parameter reduction for hyper-networks in multi-tasking architecture and parameter reduction for adapters used in HVTs.

Specifically, to reduce the parameter usage of the multi-task architecture, we decompose a hyper-network into a pair of separate hyper-networks. Unlike the existing approach, where a relatively large hyper-network is used to produce long vector that can be reshaped into the weight matrix of adapter, our decomposed hyper-networks individually produce two low-rank matrices that are multiplied to construct the adapter weights. As a result, we can rely on this low-rank approximation to reduce the parameter usage in the hyper-network yet maintain its performance on downstream tasks. In addition, to enable the hypernetworks shared across layers in HVTs, we factorize an adapter weight matrix into two kernels, including Template Kernels and Scaling Kernels. These two kernels are multiplied via Kronecker Product to fit in with different sizes of adapters, and this is achieved by controlling the sizes of Scaling Kernels based on the scaling of the layer/adapter (and using the same size of Template Kernels across layers). In this way, the parameters of adapter weights can be effectively reduced with a minimal sacrifice on the accuracy of downstream tasks.

To benchmark the problem, we construct a unified framework with the same implementation details and provide a comprehensive and fair comparison between existing parameter-efficient adaptation works in NLP on our multi-tasking dense vision problems. We also demonstrate that, with the integration of our proposed *Decomposed HyperNetworks* and *Layer-wise Scaling Kernels*, we can achieve a much better trade-off between trainable parameters and accuracies compared to the existing methods. Specifically, most of existing methods struggled to match the performance of the simple baseline, which individually fine-tunes the entire network for each task, while our method achieves better results than the simple baseline while only training less than $10\%$ of the parameters in a model. Compared with the state-of-the-art multi-tasking parameter-efficient adaptation method, Hyperformer [16], our method achieves competitive performance improvement with $\sim 90\%$ reduction in the trainable parameters of their method. Interestingly, we also observed that our proposed method brings high-performance improvement when applied to the network pre-trained on the larger dataset (ImageNet-22k). We will publicly release our code to facilitate future research.

To sum up, we list our contributions as follows:

- To the best of our knowledge, we are the first to address parameter-efficient **multi-task** adaptation for vision tasks. We develop a unified framework to benchmark several parameter-efficient fine-tuning NLP methods on dense vision tasks.

- We propose a novel method — *Polyhistor-Lite* that achieves significant performance improvement with very low training parameters compared to existing methods.

- We observe that our method can bring further performance improvements when applied to models with larger pre-trained dataset or with larger backbones.

## 2   Related Works

Parameter-efficient Learning aims to adapt a pre-trained model to a new task by only training a small number of parameters. The most straightforward method is to freeze the pre-trained encoder and only fine-tune the last layer, while, in terms of the accuracy of downstream tasks, it is still far from full fine-tuning. Thus, to achieve a better trade-off between accuracy and the number of tunable parameters, several works [7, 10, 11, 12, 16, 9, 8] have proposed more parameter-efficient methods, and we summarize these works in the following paragraphs.

**Single-Task Parameter-efficient Adaptation.** Several works build upon the Adapter [11], which is a bottleneck-like module that is placed across the architecture and trained while the rest of the original model is frozen. By changing the dimension of hidden vectors, one can easily control the trade-off between trainable parameters and accuracy. For example, Houlsby *et al.* [11] proposes to apply two adapter modules placed after the attention layers and the MLP layers respectively, while Pfeiffer *et al.* [15] only use adapters after MLP layers and show more parameter-efficiency. Furthermore, PHM-Layer [12] learns two types of matrices, one "slow" matrix shared across layers and the other "fast" matrix learned individually in different layers, to produce the adapter weight via Kronecker Product [17]. Compacter [12] further reduces the parameters by decomposing the slow matrix into two rank-one vectors. Different from their goal of sharing slow matrix across layers, we apply Kronecker Product to efficiently scale up adapters to different layer scales.

In addition, there are other parameter-efficient learning works. BitFit [7] shows simply tuning biases in all layers improves against the linear probing. Some other works fine-tune learnable vectors, such as learnable vectors in input word embeddings [18] and learnable vectors integrated with keys/values in each layer of transformers [19]. LoRA [10] produces two low-rank matrices, which are multiplied and served as a residual of attention weight matrices. While the above methods show favorable results with using fewer trainable parameters, the goal of these works is single-task adaptation.

**Multi-Task Parameter-efficient Adaptation.** When multiple tasks are learned jointly, one can share some homogeneous information across different tasks and save the parameter usage by removing the duplicated learned features. To this end, Hyperformer [16] introduces a hyper-network, which takes as input task embeddings and produces the weights of adapters in different tasks. Since only the parameters in the hyper-network need to be trained, the number of trainable parameters in task-wise components can thus be reduced in the multi-tasking setup. On the other hand, Sung *et al.* [20] shows that simply adding a single adapter on language transformer and sharing the adapter across tasks can achieve promising results in vision-and-language cross-modal tasks (*e.g.,* image captioning).

**Parameter-efficient Adaptation for Vision.** Despite the promising results, most parameter-efficient learning methods are evaluated on language transformers and NLP benchmarks, and parameter-efficient learning on Vision Transformer [9] is still an under-explored topic. A recent work, Visual Prompt Tuning (VPT) [9], initiates the study of parameter-efficient learning on Vision Transformers, and it follows the idea of prompt tuning in language tasks and prepends and fine-tunes some extra learnable vectors in the input space of pre-trained Vision Transformers. VPT focuses on single-task adaptation, while our work focuses on multi-task adaptation.

To fairly compare different parameter-efficient learning methods, He *et al.* [21] present an empirical study and re-evaluate parameter-efficient learning methods (BitFit, Adapters, Prefix Tuning, and LoRA) under the same experiment configurations for NLP tasks. Inspired by their work, we implement the aforementioned parameter-efficient NLP methods (and include more latest works [12, 16, 9]) on our dense vision tasks, conduct comparative experiments, and fairly compare these methods.

## 3   Background

**Hierarchical Vision Transformers.** The Vision Transformer [22] is based on transformer architectures [22] and operates on a set of patch tokens obtained from an image. As a variant of Vision Transformer, the Hierarchical Vision Transformer [13, 14, 23, 24, 25, 26] produces multi-scale feature representations, and its hierarchical structure extracts fine-grained information and better handles

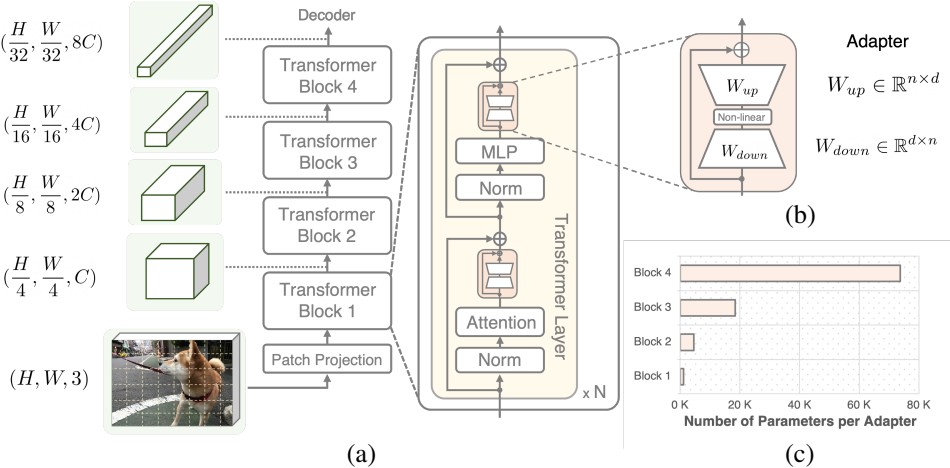

(a)             (b)             (c)

Figure 1: Illustration of (a) Hierarchical Vision Transformer and (b) Adapter. (c) When applying adapters in a Hierarchical Vision Transformer, the number of parameters grows quadratically with the respect to the block scale. Note that $C$ indicates the dimension of adapter input vectors, $n$ is the bottleneck size of adapters, and $d$ represents the input size of adapters.

images with scale and size variation. These properties contribute to the promising results in several per-pixel vision tasks, including semantic segmentation [13, 14, 26], depth estimation [27], and saliency detection [28]. As shown in Fig. 1a, a Hierarchy Vision Transformer (HVT) consists of several transformer layers, and each transformer layer is mainly composed of an attention layer and an MLP layer. Different from other transformers (*e.g.,* ViT [22]), a distinct characteristic of HVTs is its pyramid-like feature maps generated from different transformer blocks as shown in Fig. 1a.

**Adapters.** Several parameter-efficient adaptation works [11, 15, 12, 21] build upon Adapter [11], which is a bottleneck-like module placed in transformer layers as shown in Fig. 1b. These layers are learnable parameters, while the rest of the model is frozen during fine-tuning. The Adapter $f_a(\cdot)$ consists of a down-projection layer $W_{down} \in \mathbb{R}^{d \times n}$, a non-linearity function $\delta(\cdot)$, a up-projection layer $W_{up} \in \mathbb{R}^{n \times d}$, and a skip connection from the input of the adapter $h_{in} \in \mathbb{R}^d$.

$$h_{out} = f_a(h_{in}; W) = \delta(h_{in}W_{down})W_{up} + h_{in}, \tag{1}$$

where $h_{out} \in \mathbb{R}^d$ is the output of the adapter and $W = [W_{down}; W_{up}^\mathsf{T}] \in \mathbb{R}^{d \times 2n}$ represents all learnable parameters in the adapter.

# 4 Method

**Problem Setting.** Given a Hierarchical Vision Transformer pre-trained on large-scale image datasets (*e.g.,* ImageNet [29]), our goal is to train a small number of parameters and adapt the model to the multi-tasking setting, where training data of $N$ tasks are obtained during the training stage. Following the existing works in NLP, the criteria of parameter-efficient multi-tasking learning includes the accuracy of downstream tasks and the numbers of training parameters.

**Method Overview.** We aim to improve the parameter efficiency in two aspects: (1) to efficiently share homogeneous information across tasks via lightweight hyper-networks (Section 4.1) and (2) to efficiently scale up adapter weights in different transformer blocks of Hierarchical Vision Transformers (Section 4.2). These two components are combined to improve the trade-off between accuracy and training parameters in multi-tasking per-pixel vision tasks (Section 4.3).

## 4.1 *Polyhistor*: Decomposed Lightweight Hyper-networks for Multi-task Adaptation

With the goal of jointly adapting multiple NLP tasks in a parameter-efficient manner, a prior work, Hyperformer [16], builds upon a group of adapters in different tasks and extracts task-sharing information via a hyper-network shared across different tasks. Specifically, a group of task and layer-wise adapters with weight parameters $\{W_l^t | t = 1, ..., N; l = 1, ..., L\}$ are individually inserted into each layer $l$ of the model with $L$ layers for all $N$ tasks. Then, instead of individually learning the

weights of these adapters via backpropagation, Hyperformer constructs a layer-wise hyper-network $\hat{W}_l$, which takes as input a learnable task embedding $V_t$ and produce the weights of adapters $W_l^t$.

$$W_l^t = \Pi(V_t\hat{W}_l) \in \mathbb{R}^{d \times 2n},$$
$$V_t \in \mathbb{R}^k, \quad \hat{W}_l \in \mathbb{R}^{k \times 2dn}, \quad \Pi(\cdot) : \mathbb{R}^{2dn} \to \mathbb{R}^{d \times 2n} \tag{2}$$

where $\Pi(\cdot)$ maps a vector with size of $2dn$ to a matrix with size of $d \times 2n$.

While Hyperformer has shown promising results on multi-task NLP benchmarks, its effectiveness on vision tasks is unclear. In addition, since the hyper-network produces the vectorization of an adapter weight matrix (*i.e.,* $W_l^t \in \mathbb{R}^{d \times 2n}$), the output dimension of the hyper-network is the order of $\mathcal{O}(dn)$ and the size of the hyper-network becomes the order of $\mathcal{O}(dnk)$, where $d$ and $n$ are dimensions of the input and bottleneck vectors and $k$ is the size of task embeddings. For dense vision tasks, the size of input vectors is usually large (*e.g.,* $1024$ in SwinTransformer-Base). When the bottleneck dimension is set to be proportional to the input dimension (*e.g.,* $n = \frac{d}{\alpha}$, where $\alpha > 1$ is a constant), the size of hyper-network is then quadratically increased with respect to input vectors $\mathcal{O}(kd^2)$.

To alleviate this issue, we propose to decompose a single hyper-network $\hat{W}_l$ into a pair of lightweight hyper-networks $\{\hat{W}_l^p, \hat{W}_l^q\}$ , each of which only produces a low-rank matrix. We then multiply the matrices to obtain the adapter weight as shown in the top of Fig. 2a.

$$W_l^t = \sum_{i=1}^r p_i q_i^\intercal = \Pi_p(V_t\hat{W}_l^p)\,\Pi_q(V_t\hat{W}_l^q)^\intercal,$$
$$\Pi_p(V_t\hat{W}_l^p) \in \mathbb{R}^{d \times r}, \quad \Pi_q(V_t\hat{W}_l^q) \in \mathbb{R}^{2n \times r},$$
$$V_t \in \mathbb{R}^k, \quad \hat{W}_l^p \in \mathbb{R}^{k \times dr}, \quad \hat{W}_l^q \in \mathbb{R}^{k \times 2nr}, \tag{3}$$
$$\Pi_p(\cdot) : \mathbb{R}^{dr} \to \mathbb{R}^{d \times r}, \quad \Pi_q(\cdot) : \mathbb{R}^{2nr} \to \mathbb{R}^{2n \times r},$$

where $\Pi_p(\cdot)$ and $\Pi_q(\cdot)$ are matrix reshape functions and $r$ is a matrix rank which is hyper-parameter tuned according the computational budget. Note that the matrix rank $r$ is usually much smaller than dimensions of an adapter $n$ or $d$ (*i.e.,* $r << n < d$).

In this way, with the low-rank decomposition and approximation, a heavy hyper-network $\hat{W}_l \in \mathbb{R}^{k \times 2dn}$ can be reduced to two lightweight hyper-networks $\{\hat{W}_l^p \in \mathbb{R}^{k \times dr}, \hat{W}_l^q \in \mathbb{R}^{k \times 2nr}\}$, and the number of trainable parameters can be reduced from $2kdn$ to $k(d + 2n)$. The number of trainable parameters in hyper-networks is reduced from quadratic to linear increasing with respect to the input size (*i.e.,* $\mathcal{O}(kd^2) \to \mathcal{O}(kd)$). We will discuss how this translate into practical usage and demonstrate that our method can significantly save the number of training parameters while achieving competitive performance compared to the vanilla Hyperformer in Sec. 5.

### 4.2 Layer-Wise Scaling Kernels for Hierarchical Vision Transformers

Although the number of learnable parameters is reduced by sharing two lightweight hyper-networks across tasks, each layer of transformer still requires a pair of layer-wise hyper-networks. To prevent the number of parameters from growing linearly with respect to the number of layers in a transformer, one can share a hyper-network across not only different tasks but also different layers (similar to the recently introduced Hyperformer++ [16] in NLP). However, since adapters in different blocks of Hierarchical Vision Transformers have different dimensions, such a property restricts us from using the same pair of hyper-networks to produce the weights of adapters in different transformer layers.

To overcome this issue, we introduce Layer-wise Scaling Kernels to enable sharing a hyper-network across layers. To be more specific, as shown in Fig. 1, Hierarchical Vision Transformers have four transformers blocks, and each transformers block (indexed by $b$) has multiple transformer layers with a block-wise scale $s_b = 2^{b-1}$ and a output size $(\frac{H}{4s_b}, \frac{W}{4s_b}, s_bC)$. However, the increasing channel sizes in transformer blocks cause two problems. First, as shown in Fig. 1, the sizes of adapter weights increase quadratically with respect to the channel size/input size. As a result, the adapters in the later blocks would require much more trainable parameters than the earlier blocks. Second, as mentioned above, due to the different sizes of adapters in different transformer layers, a single hyper-network cannot produce multiple adapters in different layers and thus cannot be directly shared across layers.

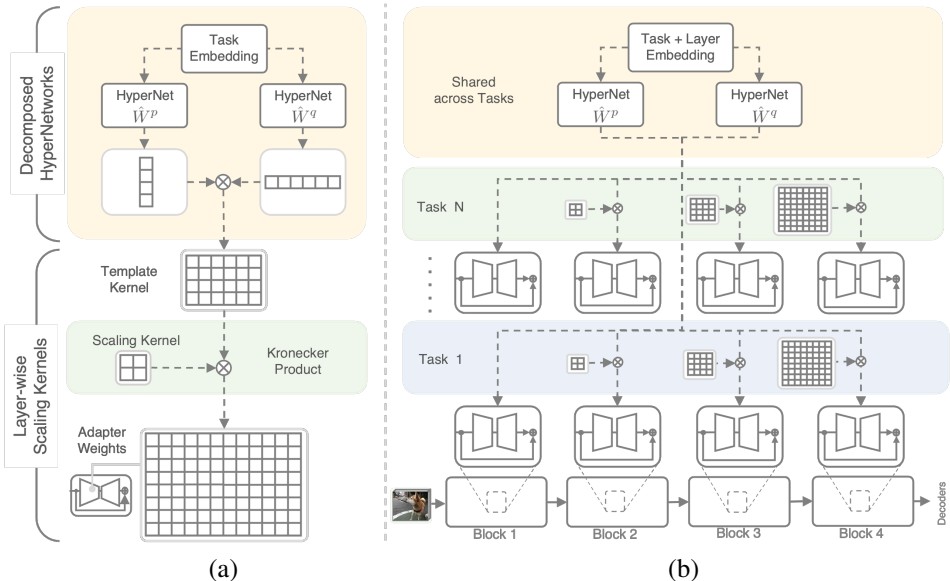

(a)                                        (b)

Figure 2: Illustration of our **Polyhistor** and **Polyhistor-Lite**. (a) We propose **Polyhistor**, which applies Decomposed HyperNetworks to reduce the number of training parameters in multi-task adaptation (Section 4.1). We also introduce Layer-wise Scaling Kernels to efficiently scale up Template Kernels for different scales of adapters (Section 4.2). (b) By combining Decomposed HyperNetowrks and Layer-wise Scaling Kernels, our **Polyhistor-Lite** can efficiently address multi-task adaptation in per-pixel vision tasks (Section 4.3).

As shown in the bottom of Figure 2a, to address these two problems, we propose to factorize the weight matrix of a adapter $W_l$ of the layer $l$ into a set of Template Kernels $\tilde{W}_l^i$ and Scaling Kernels $\kappa_l^i$. To efficiently scale up adapter weights in different transformer layers, we resort to Hypercomplex multiplication [17] and use Kronecker Product to integrate these types of matrices.[1]

$$W_l = \sum_{i=1}^{s_{\psi(l)}} \tilde{W}_l^i \otimes \kappa_l^i,$$

$$\tilde{W}_l^i \in \mathbb{R}^{d \times 2n}, \quad \kappa_l^i \in \mathbb{R}^{s_{\psi(l)} \times s_{\psi(l)}}, \quad W_l \in \mathbb{R}^{ds_{\psi(l)} \times 2ns_{\psi(l)}}, \tag{4}$$

where $s_{\psi(l)}$ and $\psi(l)$ are the scale and the index of transformer block where transformer layer $l$ is located, and $\otimes$ is Kronecker Product matrix operation. The sizes of Template Kernels are the same across layers, while the sizes of Scaling Kernels depend on the block-wise scale $s_{\psi(l)}$.

In other words, the purpose of Scaling Kernels is to scale up the Template Kernels and fit them into the adapters in layers with different scales. This decomposition not only reduces the parameter usage in a single adapter but also shows the potential to reduce parameters with a shared hyper-network.

### 4.3  *Polyhistor-Lite*: Lightweight Hypernetworks for Hierarchical Vision Transformers

We have described how to reduce the training parameters for multi-tasking adaptation in Section 4.1 and Hierarchical Vision Transformers in Section 4.2. To perform parameter-efficient adaptation for multi-tasking per-pixel vision tasks, we integrate these two components to obtain our final framework.

Specifically, as shown in Fig. 2b, we use a single pair of hyper-networks $\{\hat{W}^p, \hat{W}^q\}$ shared across different layers and different tasks. For the task $t$, the hyper-networks take as input a set of trainable layer embeddings $\{\tilde{V}_l^i\}_{i=1}^{s_{\psi(l)}}$ and a task embedding $\tilde{V}_t$ and produces two low-rank matrices, which

---

[1]For the ease of understanding, we omit the task index $t$.

are multiplied and derive a set of Template Kernels of adapters in different layers and tasks.

$$\tilde{W}_l^{t,i} = \Pi_p([\tilde{V}_t; \tilde{V}_l^i]\hat{W}^p) \, \Pi_q([\tilde{V}_t; \tilde{V}_l^i]\hat{W}^q)^\intercal, \forall i = 1, ..., \psi(l)$$
$$\Pi_p([\tilde{V}_t; \tilde{V}_l^i]\hat{W}^p) \in \mathbb{R}^{d \times r}, \quad \Pi_q([\tilde{V}_t; \tilde{V}_l^i]\hat{W}^q) \in \mathbb{R}^{2n \times r},$$
$$\tilde{V}_t, \tilde{V}_l^i \in \mathbb{R}^{\frac{k}{2}}, \quad \hat{W}^p \in \mathbb{R}^{k \times dr}, \quad \hat{W}^q \in \mathbb{R}^{k \times 2nr},$$
$$\Pi_p(\cdot) : \mathbb{R}^{dr} \to \mathbb{R}^{d \times r}, \quad \Pi_q(\cdot) : \mathbb{R}^{2nr} \to \mathbb{R}^{2n \times r}. \tag{5}$$

To derive the parameters of an adapter in each layer, we learn another set of Scaling Kernels and combine them with the Template Kernels via Kronecker Product.

$$W_{l,}^t = \sum_{i=1}^{s_{\psi(l)}} \tilde{W}_l^{t,i} \otimes \kappa_l^{t,i}, \forall t = 1, ..., T \tag{6}$$
$$\tilde{W}_l^{t,i} \in \mathbb{R}^{d \times 2n}, \quad \kappa_l^{t,i} \in \mathbb{R}^{s_{\psi(l)} \times s_{\psi(l)}}, \quad W_l^t \in \mathbb{R}^{d s_{\psi(l)} \times 2n s_{\psi(l)}}.$$

With the integration of Lightweight HyperNetworks and Layer-wise Scaling Kernels, our framework can efficiently reduce the trainable parameters in multi-task adaptation for dense vision tasks. We provide two variants of our method. *Polyhistor* solely uses the Decomposed HyperNetworks, and *Polyhistor-Lite* combines both Decomposed HyperNetworks and Layer-Wise Scaling Kernels.

## 5 Experiments

### 5.1 Implementation Details

**Dataset.** We follow prior works [30, 31] on multi-task learning for dense prediction tasks and consider PASCAL-Context [32] to construct our multi-task efficient adaptation for per-pixel benchmark. We evaluate all methods on four per-pixel tasks, 21-class semantic segmentation, 7-class human part segmentation, surface normals estimation, and saliency detection. Our evaluation metrics include the mean intersection-over-union (mIoU) for semantic segmentation, human part segmentation, and saliency detection and the mean error (mErr) for surface normals estimation.

**Model Architecture.** For the encoder, we use Swin-Transformer [13] due to its strong performance in different vision tasks and the popularity in the vision community. Our decoders for different dense tasks are based on the All-MLP decoder of Segformer [14], which uses simple linear layers and bilinear upsampling layer to efficiently perform dense vision tasks, and we adapt the number of output dimension to different tasks.

**Training.** To train our model, we use the commonly-used losses for each task. Specifically, we use the standard per-pixel cross-entropy for semantic segmentation and human part segmentation, L1 loss for surface normals estimation, and balanced cross-entropy for saliency detection. For a fair comparison, we experiment on a unified codebase implementation with the same loss functions and training iterations for all baselines and our method.

### 5.2 Baselines

**Single-task full fine-tuning** uses an individual pretrained model for each task, and **Fine-tuning decoders** freezes the feature backbone and only fine-tunes task-wise decoders for different tasks.

For single-task adaptation methods (Bitfit, VPT, PHM-Layer, Compacter, Compacter++, Adapter, Low-rank Adapter, and LoRA), we place task-wise modules for each task.

**Bitfit** [7] tune the biases in all layers, and, specifically for Swin-Transformer, and we also tune biases in patch merging layers and patch projection layers.

**VPT** [9] inserts tunable embeddings in the first input layer (VPT-shallow) and all layers (VPT-deep), and we select the best hyper-parameter (*i.e.,* 50 embeddings per layer) for all results.

**PHM layer** [12] shares a slow matrix for all layers and learns a fast matrix for each layer and place modules after attention and MLP layers, **Compacter** [12] further decomposes the fast matrix into two low-rank vectors, and **Compacter++** [12] only places modules after MLP layers.

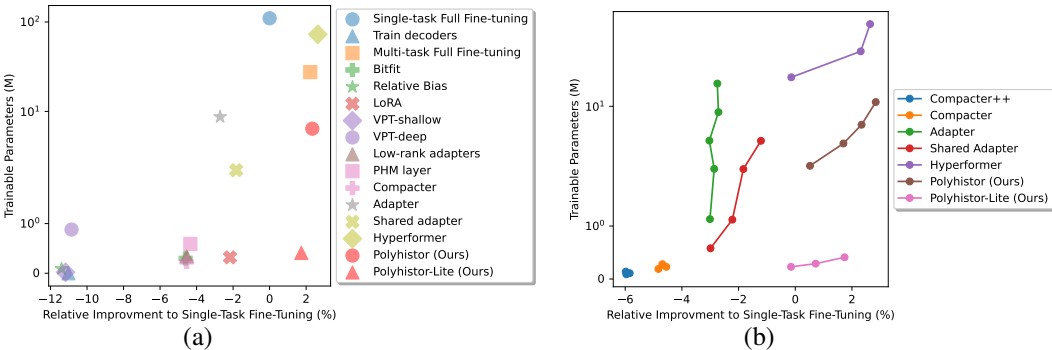

Figure 3: (a) Our *Polyhistor* uses less than one-tenth of the state-of-the-art multi-task adaptation method (*i.e.,* Hyperformer [16]) in terms of trainable parameters in the encoder. (b) Tuning hyper-parameters on the baseline methods leads to limited improvements, and we achieve the best trade-off between the trainable parameters and accuracy on downstream tasks. Details are listed in Appendix.

**LoRA** [10] applied the low-rank decomposition on attention layers, and we select rank $r = 4$ and the adapter output scale (*i.e.,* 4), which performs the best.

**Adapter** [11, 21] placed task-wise bottleneck-like modules into transformer layers, and **Shared-Adapter** [20] share an adapter across different tasks.

**Hyperformer** [16] applied a hyper-network and produce the weights for adapter, and we present the results with different adapter bottleneck dimensions. Since adapters in different layers have different dimensions, Hyperfomer++ cannot be simply adapted to Hierarchical Vision Transformers.

## 5.3 Experiment results on Multi-Task Adaptation

We evaluate all methods by computing relative improvements against Single-Task Full Fine-tuning and averaging across four tasks. Since all methods use and train all parameters of task-wise decoders, we provide two values of trainable parameters, one for the encoder and the other for the whole model.

As presented in Fig. 3 and Table 1, among all experimental methods, Hyperformer performs the best and achieves $+2.64\%$ on average for the four downstream tasks, but it requires 72M trainable parameters in the encoder. On the other hand, our *Polyhistor* achieves competitive results ($+2.34\%$), while we only need 6.41M trainable parameters in the encoder, which is less than one-tenth of the Hyperformer. Our *Polyhistor-Lite* can further reduce the trainable parameters to 0.41M by integrating the Layer-wise Scaling Kernels and sharing the hypernetwork across layers, and it achieves $+1.74\%$ and is higher than all other methods using a similar amount of trainable parameters (*e.g.,* BitFit, VPT, Shared Adapter, PHM layer, Compacter, LoRA, and Low-rank Adapter).

We also found that, while the prior parameter-efficient adaptation vision method, VPT, presented promising results on single-task image classification [9], it does not show significant improvements against the baseline that only fine-tuning decoders in our multi-task dense vision benchmark. A potential reason is that, compared to the image classification benchmarks which focus more on the same task with input shifts, our benchmark focuses more on different task outputs. This makes VPT, which adds the learnable parameters in the input space, unable to address the difference in the output space and adapt to the new tasks.

While Hyperformer achieves the best performance improvement against Single-Task Full Fine-tuning, it requires a larger number of parameters than other methods. This leads to a natural question: *Does reducing the parameters in Hyperformer derive a better trade-off between the number of tunable parameters and performance improvements?* Therefore, we decrease its size and examine whether it can maintain its performance with fewer trainable parameters. We also did a similar experiment on the other existing methods, (*e.g.,* Adapter, Shared Adapter, Compacter, and Compacter++), in which we increase the number of tuning parameters by increasing the dimension of the hidden vector in the adapter modules (increasing the down-projection ratio). As shown in Fig. 3b, we find that simply tuning the hyper-parameters in baseline methods cannot obtain a better tradeoff between the number of trainable parameters and performance improvement. For example, when the number of tuning parameters in the encoder of Hyperformer is reduced to 20.15M, its averaged relative improvement

Table 1: Experimental results on Multi-Task Adaptation. We use SwinTransformer-Tiny as the feature backbone. $\Delta_{up}$ represents relative improvement or gap to the Single-task Full Fine-tuning. Results with the symbol $\uparrow$ / $\downarrow$ indicate higher/lower is better.

| | Number of Trainable Parameters Encoder/All | Performance of Each Downstream Task | | | | Averaged Results |
|---|---|---|---|---|---|---|
| | | Seg. $\uparrow$ | H.Part $\uparrow$ | Sal. $\uparrow$ | Normals $\downarrow$ | $\Delta_{up}$ |
| Single-task Full Fine-tuning | 110.07 / 112.62 | 67.21 | 61.93 | 62.35 | 17.97 | 0.00% |
| Fine-tuning Decoders | 0.00 / 2.55 | 63.14 | 52.37 | 58.39 | 20.89 | -11.02% |
| Multi-task Full Fine-tuning | 27.51 / 30.06 | 68.71 | 62.13 | 64.18 | 17.35 | 2.23% |
| Bitfit [7] | 0.30 / 2.85 | 68.57 | 55.99 | 60.64 | 19.42 | -4.60% |
| Relative bias [13] | 0.09 / 2.64 | 63.51 | 52.35 | 57.74 | 21.07 | -11.40% |
| VPT-shallow [9] | 0.02 / 2.57 | 62.96 | 52.27 | 58.31 | 20.90 | -11.18% |
| VPT-deep [9] | 0.88 / 3.43 | 64.35 | 52.54 | 58.15 | 21.07 | -10.85% |
| PHM layer [12] | 0.59 / 3.14 | 68.55 | 56.28 | 60.35 | 19.23 | -4.34% |
| Compacter [12] | 0.23 / 2.78 | 68.08 | 56.41 | 60.08 | 19.22 | -4.55% |
| Compacter++ [12] | 0.11 / 2.66 | 67.26 | 55.69 | 59.47 | 19.54 | -5.84% |
| LoRA [10] | 0.32 / 2.87 | 70.12 | 57.73 | 61.90 | 18.96 | -2.17% |
| Adapter [21] | 8.69 / 11.24 | 69.21 | 57.38 | 61.28 | 18.83 | -2.71% |
| Low-rank adapter | 0.34 / 2.89 | 68.31 | 56.53 | 60.29 | 19.36 | -4.54% |
| Shared Adapter [20] | 2.20 / 4.74 | 70.21 | **59.15** | 62.29 | 19.26 | -1.83% |
| Hyperformer [16] | **72.77 / 75.32** | **71.43** | **60.73** | **65.54** | **17.77** | **2.64**% |
| *Polyhistor* (**Ours**) | 6.41 / 8.96 | **70.87** | **59.54** | **65.47** | **17.47** | **2.34**% |
| *Polyhistor-Lite* (**Ours**) | 0.41 / 2.96 | **70.24** | **59.12** | **64.75** | **17.40** | **1.74**% |

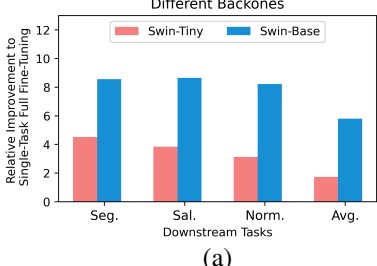

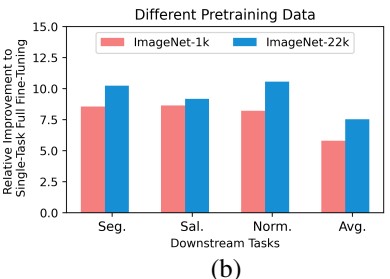

(a)                  (b)

Figure 4: Using the feature backbone with (a) larger size or (b) more pretraining data leads to larger improvements against Single-task Full Fine-tuning. All results are produced by our *Polyhistor-Lite*. For a fair comparison, both the backbones shown in (a) are pretrained on the ImageNet-1k, and both the backbones shown in (b) are based on the SwinTrasformer-Base.

shrinks to only $-0.14\%$. These experiments show that our methods achieve a better performance improvement with a fewer number of parameters than the baseline methods.

## 5.4 Ablation studies and analyses

**Different feature backbones.** To verify that our proposed method is also applicable to other larger-size model architecture, we also experiment SwinTransformer-Base. For a fair comparison, we use the same pretraining dataset, ImageNet-1k [29], on both SwinTransformer-Tiny and SwinTransformer-Base. As shown in Fig. 4a, we find that our method achieves a larger improvement against Single-task Full Fine-tuning when a larger feature backbone is used, and this shows a potential of obtaining more improvements when applying our method to a larger model architecture.

**Different Pretraining data.** We also examine how our proposed method performs when different pretraining data are used. Specifically, we apply our method to SwinTransformer-Base pretrained with ImageNet-1k and ImageNet-22k. As demonstrated in Fig. 4b, we find that our method obtains a larger performance improvement when more pretraining data was used, and this shows a potential of deriving more improvements by using more pretraining data.

**Varying trainable parameters with different pretraining data.** In Fig. 4a, we showed that using the model with more pretraining data (*i.e.,* ImageNet-22k) data can lead to a higher performance gain compared to the model with less pretraining data (*i.e.,* ImageNet-1k). To investigate this phenomenon,

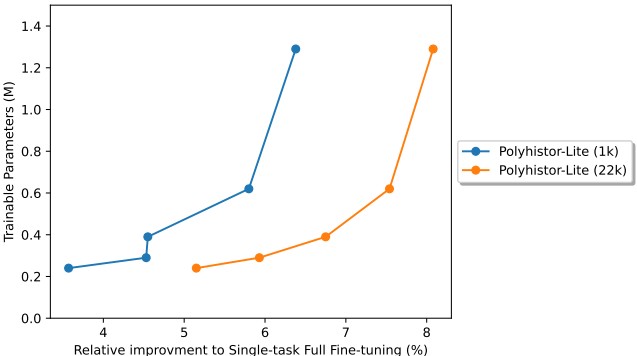

Figure 5: Our *Polyhistor-Lite* using more pretraining data can lead to higher relative improvements to the Single-task Fine-tuning baseline, and this trend is consistent across varying amounts of trainable parameters. Note that Single-task Fine-tuning baselines for *Polyhistor-Lite*(1k) and *Polyhistor-Lite*(22k) are trained on ImageNet-1k and ImageNet-22k models respectively.

we vary the size of our *Polyhistor-Lite* and measure the performance gain of the models with different pretraining data.

As shown in Fig. 5, we find that the model pretrained on ImageNet-22k can use less trainable parameters to achieve similar performance gains compared to the model pretrained on ImageNet-1k. In addition, under the same amount of trainable parameters, the model pretrained on ImageNet-22k can consistently outperform the model pretrained on ImageNet-1k. This suggests that, with more pretraining data, feature extractors can learn more diverse representations, so that we can use less trainable parameters and adapt better to different downstream tasks.

**Dimensions of task embeddings.** In addition, we also conduct analysis to examine how our method performs with different sizes of task embeddings. As presented in Table 2, when large sizes of task embeddings are used, the averaged improvement against single-task fine-tuning becomes larger.

Table 2: Ablation study on the sizes of task embeddings. We vary the sizes of task embeddings $k$ from 16 to 64 on our *Polyhistor-Lite*. All results in this table are based on SwinTransformer-Tiny pretrained on ImageNet-1k.

|  | Size of Task Embeddings $k$ | Num. of Trainable Parameters Encoder / All | Seg. ↑ | H.Seg. ↑ | Sal. ↑ | Normals ↓ | Averaged Results $\Delta_{up}$ |
|---|---|---|---|---|---|---|---|
| Single-task Full Fine-tuning | - | 110.07 / 112.62 | 67.21 | 61.93 | 62.35 | 17.97 | 0.00% |
| *Polyhistor-Lite* | 16 | 0.23 / 2.78 | 69.67 | 58.38 | 63.55 | 18.05 | -0.15% |
| *Polyhistor-Lite* | 32 | 0.29 / 2.84 | 69.80 | 58.72 | 64.14 | 17.73 | 0.72% |
| *Polyhistor-Lite* | 64 | 0.41 / 2.96 | 70.24 | 59.12 | 64.75 | 17.40 | 1.74% |

We present more ablation studies, analyses, and implementation details in Appendix.

# 6 Conclusion

We proposed *Polyhistor* and *Polyhistor-Lite* — parameter-efficient fine-tuning methods for dense vision tasks. We showed that most of the existing parameter-efficient single-task adaptation methods achieved lower performance compared with Single-task Full Fine-tuning, and the state-of-the-art multi-task adaptation method achieve favorable results while using a large number of tunable parameters. Compared to these existing approaches, our proposed methods do not only achieve a competitive performance gain to the state-of-the-art but also only use a very limited amount of tunable parameters. The potential limitation of our method is searching for suitable hyper-parameters, which is a common limitation among all parameter-efficient learning methods.

**Acknowledgments.** Yen-Cheng Liu and Zsolt Kira were partly supported by DARPA's Learning with Less Labels (LwLL) program under agreement HR0011-18-S-0044, as part of their affiliation with Georgia Tech.

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
