# Polyhistor: Parameter-Efficient Multi-Task Adaptation for Dense Vision Tasks Appendix

# 1 Additional analyses

## 1.1 Feature backbones with different pretraining data.

**Comparison to baselines.** We also experiment with other baseline methods on SwinTransformer-Base pretrained on ImageNet-1k and ImageNet-22k. As shown in Table 1 and 2, we find that the trend of all methods are similar to the results on SwinTransformer-Tiny. Specifically, most baseline methods have a performance gap with Hyperformer and our *Polyhistor-Lite*. Compared to Hyperformer, our *Polyhistor-Lite* use less than $5\%$ of their trainable parameters in the encoder to achieve similar or even better results. By tuning the adapter down-projection ratio and using fewer trainable parameters, our *Polyhistor-Lite* still obtains a higher performance gain against the baseline methods with low trainable parameters (*e.g.,* Compacter++, Bitfit, VPT, PHM layer, LoRA, Adapter). These results show that our proposed *Polyhistor-Lite* can derive a better trade-off against all existing parameter-efficient works in different pretrained feature backbones.

## 1.2 Ranks of hyper-network output.

Another important hyper-parameter in our model is ranks of hyper-network outputs. We thus experiment *Polyhistor* and vary the rank of output matrices, and, as shown in Table 3, we find the model can derive a better results with a large rank.

## 1.3 Experiment results of tuning baseline methods.

In Fig. 3b of the main paper, we presented the results of different baseline methods with different hyper-parameters. For a clearer comparison between baseline methods, We also provide the exact values of all results in Table 4. It is worth noting that simply tuning the baseline methods only leads to limited improvements of baseline methods, and our proposed *Polyhistor* and *Polyhistor-Lite* still achieve a better trade-off between performance gain and trainable parameters.

## 1.4 Experiment results of other hierarchical vision transformers.

As shown in Table 5, we further apply our method and other baseline methods to the Pyramid Vision Transformer [9]. We find our Polyhistor can achieve comparable results to Hyperformer while using much fewer trainable parameters. Polyhistor-Lite can further reduce trainable parameters and achieves higher accuracy than all other methods using a similar amount of trainable parameters (e.g., BitFit, PHM layer, Compacter, LoRA, and Low-rank Adapter). This trend is aligned with what we found in Swin Transformer. We show that our method generalizes to different backbones.

36th Conference on Neural Information Processing Systems (NeurIPS 2022).

Table 1: Experimental results on Multi-Task Adaptation. We use SwinTransformer-Base pretrained on **ImageNet-1k** as the feature backbone. $\Delta_{up}$ represents relative improvement or gap to the Single-task Full Fine-tuning. Results with the symbol $\uparrow$ / $\downarrow$ indicate higher/lower is better. $\rho$ represents the down-projection ratio of adapters (*i.e.,.* the ratio of adapter input $d$ to hidden vectors $n$, $\rho = d/n$).

| | Number of Trainable Parameters | Performance of Each Downstream Task | | | | Averaged Results |
| | Encoder/All | Seg. $\uparrow$ | H.Part $\uparrow$ | Sal. $\uparrow$ | Normals $\downarrow$ | $\Delta_{up}$ |
|---|---|---|---|---|---|---|
| Single-task Full Fine-tuning | 346.96 / 350.01 | 67.88 | 64.47 | 61.26 | 18.85 | 0.00% |
| Fine-tuning Decoders | 0.00 / 3.04 | 68.98 | 55.57 | 58.37 | 21.36 | -7.55% |
| Bitfit [1] | 0.20 / 3.24 | 71.93 | 59.12 | 60.67 | 20.08 | -2.46% |
| Relative bias [2] | 0.06 / 3.11 | 68.44 | 55.70 | 57.27 | 21.63 | -8.51% |
| VPT-deep [3] | 2.41 / 5.45 | 68.80 | 55.85 | 58.25 | 21.37 | -7.57% |
| PHM layer [4] | 1.89 / 4.94 | 71.93 | 59.11 | 59.71 | 20.35 | -3.21% |
| Compacter++ [4] | 0.25 / 3.29 | 72.00 | 59.11 | 59.73 | 20.41 | -3.25% |
| LoRA [5] | 0.87 / 4.31 | 74.10 | 61.57 | 63.87 | 18.55 | 2.63% |
| Adapter [6] | 3.64 / 6.68 | 73.29 | 60.30 | 62.42 | 18.66 | 1.10% |
| Low-rank adapter | 0.34 / 2.89 | 72.13 | 59.10 | 59.81 | 20.28 | -3.01% |
| Hyperformer [7] | **60.88 / 63.92** | **73.60** | **63.82** | **67.31** | **16.90** | **6.91**% |
| *Polyhistor-Lite* (**Ours**; $\rho = 1$) | 1.29 / 4.34 | **73.70** | **63.32** | **66.50** | **16.93** | **6.38**% |
| *Polyhistor-Lite* (**Ours**; $\rho = 2$) | 0.62 / 3.67 | **73.69** | **63.04** | **66.56** | **17.30** | **5.80**% |
| *Polyhistor-Lite* (**Ours**; $\rho = 4$) | 0.39 / 3.43 | **73.57** | **62.04** | **65.84** | **17.70** | **4.55**% |
| *Polyhistor-Lite* (**Ours**; $\rho = 8$) | 0.29 / 3.34 | **73.92** | **62.15** | **65.37** | **17.70** | **4.53**% |
| *Polyhistor-Lite* (**Ours**; $\rho = 32$) | 0.24 / 3.28 | **73.80** | **61.32** | **64.64** | **17.92** | **3.57**% |

Table 2: Experimental results on Multi-Task Adaptation. We use SwinTransformer-Base pretrained on **ImageNet-22k** as the feature backbone. $\Delta_{up}$ represents relative improvement or gap to the Single-task Full Fine-tuning. Results with the symbol $\uparrow$ / $\downarrow$ indicate higher/lower is better. $\rho$ represents the down-projection ratio of adapters (*i.e.,.* the ratio of adapter input $d$ to hidden vectors $n$, $\rho = d/n$).

| | Number of Trainable Parameters | Performance of Each Downstream Task | | | | Averaged Results |
| | Encoder/All | Seg. $\uparrow$ | H.Part $\uparrow$ | Sal. $\uparrow$ | Normals $\downarrow$ | $\Delta_{up}$ |
|---|---|---|---|---|---|---|
| Single-task Full Fine-tuning | 346.96 / 350.01 | 70.72 | 67.47 | 61.00 | 18.73 | 0.00% |
| Fine-tuning Decoders | 0.00 / 3.04 | 73.33 | 60.56 | 59.13 | 21.38 | -5.94% |
| Bitfit [1] | 0.20 / 3.24 | 76.42 | 64.89 | 62.05 | 19.03 | 1.09% |
| Relative bias [2] | 0.06 / 3.11 | 72.86 | 60.64 | 58.44 | 21.51 | -6.53% |
| VPT-deep [3] | 2.41 / 5.45 | 74.21 | 61.41 | 58.80 | 21.61 | -5.90% |
| PHM layer [4] | 1.89 / 4.94 | 76.33 | 64.59 | 60.43 | 20.23 | -1.32% |
| Compacter++ [4] | 0.25 / 3.29 | 75.99 | 64.65 | 60.42 | 20.01 | -1.13% |
| LoRA [5] | 0.87 / 4.31 | 78.24 | 66.95 | 64.70 | 18.07 | 4.86% |
| Adapter [6] | 3.64 / 6.68 | 77.22 | 65.95 | 63.80 | 18.38 | 3.35% |
| Low-rank adapter | 0.34 / 2.89 | 75.65 | 64.75 | 60.50 | 20.03 | -1.21% |
| Hyperformer [7] | **60.88 / 63.92** | **78.41** | **68.94** | **67.50** | **16.80** | **6.91**% |
| *Polyhistor-Lite* (**Ours**; $\rho = 1$) | 1.29 / 4.34 | **77.91** | **68.02** | **66.89** | **16.54** | **8.08**% |
| *Polyhistor-Lite* (**Ours**; $\rho = 32$) | 0.24 / 3.28 | **77.74** | **66.33** | **65.03** | **17.65** | **5.15**% |

## 1.5 Experiment results of self-supervised models.

We conduct an experiment using the self-supervised Swim Transformer-Tiny (MoBY-Tiny [10]), and, for a fair comparison, we also run all baseline with MoBY-Tiny and report the results in the Table 6. We find our proposed method can achieve similar or even better results compared to the Hyperformer [2] while using much less trainable parameters.

## 1.6 Discussion on difference to Visual Prompt Tuning [3]

We summarize the difference between Visual Prompt Tuning and our method in the following points.

**Different Problem Settings**: Visual Prompt Tuning focuses on single-task parameter-efficient adaptation, while our proposed method focuses on multi-task parameter-efficient adaptation. Our goal is to perform a parameter-efficient adaptation for multiple tasks and share the beneficial information across multiple vision tasks.

Table 3: Ablation study on the sizes of ranks in hypernetwork output matrices. We vary the dimensions of ranks $r$ from 1 to $\frac{n}{2}$ on our *Polyhistor*. Note that $n$ is the dimension of hidden vectors in adapters. All results in this table are based on SwinTransformer-Tiny pretrained on ImageNet-1k.

| | Dimension of Ranks $r$ | Num. of Trainable Parameters Encoder/ All | Performance of Each Downstream Task | | | | Averaged Results $\Delta_{up}$ |
|---|---|---|---|---|---|---|---|
| | | | Seg. ↑ | H.Seg. ↑ | Sal. ↑ | Normals ↓ | |
| Single-task Full Fine-tuning | - | 110.07 / 112.62 | 67.21 | 61.93 | 62.35 | 17.97 | 0.00% |
| *Polyhistor* | 1 | 2.38 / 4.93 | 70.31 | 58.61 | 64.14 | 17.98 | 0.52% |
| *Polyhistor* | n/8 | 4.08 / 6.63 | 71.18 | 59.52 | 65.04 | 17.81 | 1.70% |
| *Polyhistor* | n/4 | 6.41 / 8.96 | 70.87 | 59.54 | 65.47 | 17.47 | 2.34% |
| *Polyhistor* | n/2 | 11.08 / 13.63 | 71.31 | 60.15 | 65.46 | 17.40 | 2.84% |

Table 4: Limited improvements from tuning hyper-parameters on baseline method. $\Delta_{up}$ represents relative improvement or gap to the Single-task Full Fine-tuning. Results with the symbol ↑ / ↓ indicate higher/lower is better. Results with the symbol ↑ / ↓ indicate higher/lower is better. $\rho$ represents the down-projection ratio of adapters (*i.e.,*. the ratio of adapter input $d$ to hidden vectors $n$, $\rho = d/n$).

| | Num. of Trainable Parameters Encoder / All | Performance of Each Downstream Task | | | | Averaged Results $\Delta_{up}$ |
|---|---|---|---|---|---|---|
| | | Seg. ↑ | H.Seg. ↑ | Sal. ↑ | Normals ↓ | |
| Single-task Full Fine-tuning | 110.07 / 112.62 | 67.21 | 61.93 | 62.35 | 17.97 | 0.00% |
| Fine-tuning Decoders | 0.00 / 2.55 | 63.14 | 52.37 | 58.39 | 20.89 | -11.02% |
| Compacter++ ($\rho = 1$) [4] | 0.14 / 2.69 | 67.33 | 55.68 | 59.50 | 19.66 | -5.98% |
| Compacter++ ($\rho = 2$) [4] | 0.11 / 2.66 | 67.26 | 55.69 | 59.47 | 19.54 | -5.84% |
| Compacter++ ($\rho = 8$) [4] | 0.09 / 2.64 | 67.19 | 55.85 | 59.48 | 19.56 | -5.96% |
| Compacter ($\rho = 1$) [4] | 0.28 / 2.83 | 67.94 | 56.23 | 60.18 | 19.25 | -4.69% |
| Compacter ($\rho = 2$) [4] | 0.23 / 2.78 | 68.08 | 56.41 | 60.08 | 19.22 | -4.55% |
| Compacter ($\rho = 8$) [4] | 0.19 / 2.74 | 68.15 | 56.16 | 60.12 | 19.37 | -4.83% |
| Adapter ($\rho = 1$) [6] | 17.32 / 19.87 | 69.13 | 57.35 | 61.17 | 18.79 | -2.75% |
| Adapter ($\rho = 2$) [6] | 8.69 / 11.24 | 69.21 | 57.38 | 61.28 | 18.83 | -2.71% |
| Adapter ($\rho = 4$) [6] | 4.37 / 6.92 | 68.93 | 57.33 | 61.24 | 18.95 | -3.03% |
| Adapter ($\rho = 8$) [6] | 2.21 / 4.76 | 69.04 | 57.34 | 61.25 | 18.86 | -2.86% |
| Adapter ($\rho = 16$) [6] | 1.13 / 3.68 | 69.03 | 57.22 | 61.17 | 18.91 | -3.01% |
| Shared Adapter ($\rho = 1$) [8] | 4.35 / 6.89 | 70.57 | 59.43 | 62.54 | 19.07 | -1.21% |
| Shared Adapter ($\rho = 2$) [8] | 2.20 / 4.74 | 70.21 | 59.15 | 62.29 | 19.26 | -1.83% |
| Shared Adapter ($\rho = 4$) [8] | 1.12 / 3.66 | 70.02 | 58.87 | 62.09 | 19.35 | -2.22% |
| Shared Adapter ($\rho = 8$) [8] | 0.58 / 3.12 | 69.63 | 58.54 | 61.74 | 19.61 | -2.99% |
| Hyperformer ($\rho = 8$) [7] | 72.77 / 75.32 | 71.43 | 60.73 | 65.54 | 17.77 | 2.64% |
| Hyperformer ($\rho = 16$) [7] | 37.69 / 40.24 | 71.28 | 60.19 | 65.82 | 17.89 | 2.31% |
| Hyperformer ($\rho = 32$) [7] | 20.15 / 22.70 | 71.12 | 59.71 | 64.41 | 19.06 | -0.14% |
| *Polyhistor*(**Ours**) | 6.41 / 8.96 | 70.87 | 59.54 | 65.47 | 17.47 | 2.34% |
| *Polyhistor-Lite*(**Ours**) | 0.41 / 2.96 | 70.24 | 59.12 | 64.75 | 17.40 | 1.74% |

**Different types of parameter-efficient methods**: Visual Prompt Tuning adds learnable parameters along with the visual embeddings, while our proposed method utilizes a shared hyper-network to produce the adapter weights for different tasks. Also, the insertion locations of learnable parameters are different (VPT: input space, Ours: parallel to fully-connected layers).

## 2 Implementation Details

For a fair comparison between different methods, we use batch size 12 and train for 60 epochs for each task. We use Adam optimizer [11] with the learning rate $1e - 4$ and the weight decay $1e - 4$, and the learning rate is linearly decreased with respect to the training iteration.

We followed the prior multi-tasking learning work [12] to use task-wise weighting on different losses, while we found that using the uniform weights on the losses has similar results as the task-wise weighting. We also applied the same data augmentations, `RandomHorizontalFlip`, `RandomScale`

Table 5: Experimental results on Multi-Task Adaptation. We use Pryramid Vision Transformer-Small as the feature backbone. $\Delta_{up}$ represents relative improvement or gap to the Single-task Full Fine-tuning. Results with the symbol $\uparrow$ / $\downarrow$ indicate higher/lower is better. $\rho$ represents the down-projection ratio of adapters (*i.e.,*. the ratio of adapter input $d$ to hidden vectors $n$, $\rho = d/n$).

| | Number of Trainable Parameters Encoder/All | Performance of Each Downstream Task | | | | Averaged Results $\Delta_{up}$ |
|---|---|---|---|---|---|---|
| | | Seg. $\uparrow$ | H.Part $\uparrow$ | Sal. $\uparrow$ | Normals $\downarrow$ | |
| Single-task Full Fine-tuning | 95.88 / 97.99 | 68.81 | 61.27 | 62.67 | 17.55 | 0.00% |
| Fine-tuning Decoders | 0.00 / 2.11 | 64.86 | 51.18 | 61.54 | 19.55 | -8.85% |
| Bitfit [1] | 0.22 / 2.34 | 71.41 | 55.71 | 64.08 | 18.69 | -2.38% |
| Adapter [6] | 0.79 / 2.90 | 71.94 | 56.38 | 64.16 | 18.75 | -1.97% |
| LoRA [5] | 0.30 / 2.41 | 71.89 | 56.90 | 64.27 | 18.48 | -1.35% |
| Low-rank adapter | 0.25 / 2.36 | 70.72 | 55.34 | 63.39 | 18.70 | -3.08% |
| PHM layer [4] | 0.42 / 2.53 | 70.81 | 55.02 | 63.51 | 18.75 | -3.20% |
| Compacter++ [4] | 0.09 / 2.20 | 70.29 | 54.80 | 63.16 | 18.82 | -3.71% |
| Hyperformer [7] | **14.03 / 16.14** | **70.81** | **57.76** | **65.49** | **17.75** | **0.14%** |
| *Polyhistor-Lite* (**Ours**; $\rho = 1$) | 5.21 / 7.32 | **71.00** | **57.52** | **65.83** | **17.83** | **0.13%** |
| *Polyhistor-Lite* (**Ours**; $\rho = 32$) | 0.29 / 2.40 | **70.93** | **56.71** | **65.00** | **17.95** | **-0.73%** |

Table 6: Experimental results on Multi-Task Adaptation. We use MoBY-Tiny [10] as the feature backbone. $\Delta_{up}$ represents relative improvement or gap to the Single-task Full Fine-tuning. Results with the symbol $\uparrow$ / $\downarrow$ indicate higher/lower is better. $\rho$ represents the down-projection ratio of adapters (*i.e.,*. the ratio of adapter input $d$ to hidden vectors $n$, $\rho = d/n$).

| | Number of Trainable Parameters Encoder/All | Performance of Each Downstream Task | | | | Averaged Results $\Delta_{up}$ |
|---|---|---|---|---|---|---|
| | | Seg. $\uparrow$ | H.Part $\uparrow$ | Sal. $\uparrow$ | Normals $\downarrow$ | |
| Single-task Full Fine-tuning | 110.07 / 112.62 | 65.52 | 61.78 | 62.05 | 18.14 | 0.00% |
| Fine-tuning Decoders | 0.00 / 2.55 | 59.64 | 52.97 | 59.60 | 19.88 | -9.21% |
| Bitfit [1] | 0.30 / 2.85 | 63.43 | 54.90 | 59.50 | 19.80 | -6.90% |
| VPT-shallow [3] | 0.02 / 2.57 | 59.50 | 52.84 | 59.48 | 19.88 | -9.36% |
| VPT-deep [3] | 0.88 / 3.43 | 56.15 | 50.30 | 57.22 | 20.71 | -13.72% |
| Adapter [6] | 8.69 / 11.24 | 65.00 | 56.66 | 60.84 | 18.64 | -3.45% |
| LoRA [5] | 0.32 / 2.87 | 65.64 | 57.66 | 62.29 | 18.47 | -1.99% |
| Low-rank adapter | 0.34 / 2.89 | 63.30 | 55.24 | 59.72 | 19.14 | -5.82% |
| PHM layer [4] | 0.59 / 3.14 | 63.21 | 54.99 | 59.70 | 19.13 | -5.95% |
| Compacter++ [4] | 0.11 / 2.66 | 62.31 | 54.69 | 59.43 | 19.58 | -7.14% |
| Hyperformer [7] | **19.29 / 44.25** | **66.50** | **58.97** | **66.02** | **17.61** | **1.56%** |
| *Polyhistor* (**Ours**) | 6.41 / 8.96 | **67.69** | **59.32** | **65.15** | **17.43** | **2.05%** |
| *Polyhistor-Lite* (**Ours**) | 0.41 / 2.96 | **67.23** | **58.90** | **64.62** | **17.72** | **1.09%** |

with the range $[0.75, 1.25]$, `RandomRotate` with the range $[-20, 20]$, and `Resize` to $(512, 512)$, which are used in the prior work [12].

For the hyper-parameters of *Polyhistor*, we set the input dimension of adapter $d$ as the dimension of hidden vectors in SwinTransformers, and the down-projection ratio is set as $\rho = d/n = 16$. For the low-rank output matrix of hyper-networks, we set the rank as $n/4$, where $n$ is bottleneck size. We set the size of task embeddings as $64$.

As for the hyper-parameters of *Polyhistor-Lite*, we also set the input dimension of adapter $d$ as the dimension of hidden vectors in SwinTransformers, and the down-projection ratio is set as $\rho = d/n = 2$. For the low-rank output matrix of hyper-networks, we set the rank as $n/4$, where $n$ is bottleneck size. We set the size of task embeddings as $64$.