# OpenReview forum: "Polyhistor: Parameter-Efficient Multi-Task Adaptation for Dense Vision Tasks"
_NeurIPS.cc/2022/Conference — NeurIPS 2022 Accept_

### Official Review · Reviewer_zAUZ · 2022-07-07

**Rating:** 6
**Confidence:** 4
**Soundness:** 3 good
**Presentation:** 4 excellent
**Contribution:** 3 good

**Summary:**

The paper proposes Polyhistor, a parameter efficient tuning method for jointly tuned dense vision tasks. Previous adapter-like methods in NLP are benchmarked in detail for dense vision tasks. The proposed method is proved to give better parameter-performance trade-off on the studied tasks.

**Questions:**

I am just curious about why the four tasks are jointly tuned. The reason that we need parameter-efficient tuning is that we can not afford replicating a model with 500B parameters on 1000 down stream tasks, especially when the tasks come one by one, unexpectedly. If we jointly tune the model for a few tasks, how do we do with the upcoming tasks?

I do not mean to criticize the paper on this point. It's a good paper, and the world needs diversity. I just have this confusion, and some other readers may also have this confusion. So, maybe this is a problem worth addressing in the paper.

**Limitations:**

1. Only pretrained Swin Transformer is studied in this paper, but I feel this method can be easily extended to ConvNets.
2. More experiments on models pretrained on SSL tasks should also make this paper stronger, since SSL models are competitive against IN-pretrained models, and the features are very different. The behavior of a method can be very different depending on the pretrain task.

**Strengths And Weaknesses:**

Strengths:

1. The paper is neatly written. It is easy to follow and understand.
2. I haven't seen a lot of paper to study parameter-efficient tuning specifically for dense vision tasks, so the benchmark for adapter-like methods in this paper should be helpful for the community. The authors also promised to release the code.
3. The idea makes sense to me, and the performance is well-supported by the experiments. Low-rank decomposition and dynamic weight are nothing new, but they are properly applied. The Layer-wise Scaling Kernel is novel and effective.

Weakness:

The comparison w.r.t single task baselines doesn't seem fair to me. A model jointly trained on similar tasks is expected to outperform models trained on single tasks respectively. I feel results on single-task should be reported. It's ok to perform worse on a few tasks, since this paper is mainly comparing with multitask methods, but the results should be presented for completeness.

---

> ### Author Response · Authors · 2022-08-02
> **We added one experiment on a self-supervised model and one experiment of sequential multi-task setting.**
>
> ***Q1: Could this method be applied to self-supervised backbones?***
>
> **A:** Thanks for the suggestion. We are also curious about this interesting question and conducted an experiment using the self-supervised Swim Transformer-Tiny (MoBY-Tiny [1]). For a fair comparison, we also run all baselines with MoBY-Tiny and report the results in the following Table.
>
> |             |                       ▼ Results of **MoBY-Tiny [1]**                              |                   |                    |                  |                     |                   |
> |:----------------------------:|:------------------------------------------------:|:----------------:|:------------------:|:----------------:|:-------------------:|:-----------------:|
> |          **Methods**          | **Trainable Parameters (Encoder/All; Millions)** | **Seg.↑ (mIoU)** | **H.Seg.↑ (mIoU)** | **Sal.↑ (mIoU)** | **Normals↓ (mErr)** | **Avg. Improve.** |
> | Single-task full fine-tuning |                   110.07/112.62                  |       65.52      |        61.78       |       62.05      |        18.14        |       0.00%       |
> |     Fine-tuning Decoders     |                     0.00/2.55                    |       59.64      |        52.97       |       59.60      |        19.88        |       -9.21%      |
> |            Bitfit            |                     0.30/2.85                    |       63.43      |        54.90       |       59.50      |        19.80        |       -6.90%      |
> |          VPT-shallow         |                     0.02/2.57                    |       59.50      |        52.84       |       59.48      |        19.88        |       -9.36%      |
> |           VPT-deep           |                     0.88/3.43                    |       56.15      |        50.30       |       57.22      |        20.71        |      -13.72%      |
> |            Adapter           |                    8.69/11.24                    |       65.00      |        56.66       |       60.84      |        18.64        |       -3.45%      |
> |             LoRA             |                     0.32/2.87                    |       65.64      |        57.66       |       62.29      |        18.47        |       -1.99%      |
> |       Low-rank adapter       |                     0.34/2.89                    |       63.30      |        55.24       |       59.72      |        19.14        |       -5.82%      |
> |           PHM layer          |                     0.59/3.14                    |       63.21      |        54.99       |       59.70      |        19.13        |       -5.95%      |
> |          Compacter++         |                     0.11/2.66                    |       62.31      |        54.69       |       59.43      |        19.58        |       -7.14%      |
> |          Hyperformer         |                    19.29/44.25                   |       66.50      |        58.97       |       66.02      |        17.61        |       1.56%       |
> |        **Polyhistor**        |                   **6.41/8.96**                  |     **67.69**    |      **59.32**     |     **65.15**    |      **17.43**      |     **2.05%**     |
> |      **Polyhistor-Lite**     |                   **0.41/2.96**                  |     **67.23**    |      **58.90**     |     **64.62**    |      **17.72**      |     **1.09%**     |
>
> ---

---

> > ### Author Response · Authors · 2022-08-02
> > **Cont'd**
> >
> > ***Q2: What if the tasks come one by one?***
> >
> > **A:** We thank the reviewer for asking this question, and this is an interesting question worth exploring. Therefore, we consider the sequential tasks scenario, where each task is available at a time and four tasks sequentially appear.
> >
> > - **[Comparison between joint training and sequential training]** Since the joint training allows one to use the training data from all tasks to jointly train the hypernetwork, the hypernetwork can share information across similar tasks and thus achieve slightly higher results compared to the sequential training. However, the sequential training does not have the constraint of accessing all training data from all tasks at the same time.
> >
> > - **[Benefit of our method in sequential training scenario]** Usually, the sequential training or continual learning scenario has the forgetting issue [3, 4], where the model learned on new tasks degrades its accuracy of previous tasks. Interestingly, we find our framework does not suffer from such an issue.
> >
> >   It is because our adapter weights are generated from the hypernetwork and can be stored ***offline***, and we can easily insert the stored adapters into the pretrained feature backbone to perform the learned task (note that the pretrained model is frozen all the time). Therefore, even when the hypernetwork is trained to learn new tasks and generates adapter weights for the new tasks, the learning of new tasks will not affect the model inference of the previous tasks.
> >
> >   To be more specific, once we learned the hyper-network to generate the adapters for a specific task, we can store the adapters offline and use the trained hypernetwork to continually learn new tasks. Although the forgetting issue might appear in the  hypernetwork, the previously stored adapters will not be affected and can be inserted into the pretrained feature backbone to perform previously learned tasks.
> >
> >
> > |               |   Time  | Seg.↑ (mIoU) | H.Seg.↑ (mIoU) | Sal.↑ (mIoU) | Normals↓ (mErr) |
> > |:-------------:|:---:|:------------:|:--------------:|:------------:|:---------------:|
> > |   Sequential  | T=0 |     69.79    |        -       |       -      |        -        |
> > |               | T=1 |     69.79    |      63.99     |       -      |        -        |
> > |               | T=2 |     69.79    |      63.99     |     58.59    |        -        |
> > |               | T=3 |     69.79    |      63.99     |     58.59    |      17.87      |
> > |               |     |              |                |              |                 |
> > | Joint 4-tasks |  -   |     70.24    |      64.75     |     59.12    |      17.40      |
> >
> >
> > ---
> >
> > **Reference:**
> >
> > [1] “Self-Supervised Learning with Swin Transformers”,  Xie et al., arXiv 2021
> >
> > [2] “Parameter-efficient Multi-task Fine-tuning for Transformers via Shared Hypernetworks”, Mahabadi et al., ACL 2021
> >
> > [3] “Overcoming catastrophic forgetting in neural networks”, Kirkpatrick et al., arXiv 2016
> >
> > [4] “Learning without Forgetting”, Li et al., PAMI 2018

---

> > > ### Comment · Reviewer_zAUZ · 2022-08-08
> > > **Re: Rebuttal**
> > >
> > > Thank you for your response. I will keep my score unchanged.

---

### Official Review · Reviewer_eYWB · 2022-07-10

**Rating:** 7
**Confidence:** 4
**Soundness:** 3 good
**Presentation:** 3 good
**Contribution:** 4 excellent

**Summary:**

This paper proposes a parameter-efficient multi-task adaptation method for dense vision tasks, called Polyhistor-lite. It is used to adapt a pre-trained hierarchical vision transformer for solving multiple dense vision tasks. The proposed method consists of two aspects, the Decomposed Hyper-networks and the Layer-Wise Scaling Kernels. Models are evaluated on PASCAL-Context datasets for semantic segmentation, human part segmentation, surface normals estimation, and saliency detection and are shown to be effective.

**Questions:**


See Weaknesses point 2 for the question. Overall, this proposed method is novel and effective with well-presentation.


**Limitations:**

Limitations are discussed in the paper.

**Strengths And Weaknesses:**


**Strengths**
1. It is interesting to explore the parameter-efficient adaptation techniques for dense vision tasks. It has not been investigated in this area.
2. The presentation of this work is clear. This paper also provides a detailed discussion of the differences and relations with parameter-efficient multi-task adaptation methods in NLP tasks.
3. The proposed parameter-sharing method is reasonable and is shown to be helpful in reducing the learning parameters but also keeping relatively high performance.
4. Comparisons with existing approaches are thorough and significant.

**Weaknesses**
1. In Figure 2(b), the Transformer in the lower part has no direct connection with the upper part and is meaningless.
2. The proposed method is only evaluated with Swin-Transformer. As claimed by this paper, it is designed for hierarchical vision transformers. Would it also work well with other hierarchical vision transformers?

---

> ### Author Response · Authors · 2022-08-02
> **We added the experiment for another feature backbone and modified the figure 2**
>
> ***Q1: Would the proposed method also work well with other hierarchical vision transformers?***
>
> **A:**
> Yes, our method can be applied to other backbones. In addition to applying it to the Swin Transformer in the paper, for the rebuttal we further apply our method and other baseline methods to the **Pyramid Vision Transformer** [1] as shown in the Table. The conclusions are consistent with our original experiments. We find our Polyhistor can achieve comparable results to Hyperformer while using much fewer trainable parameters. Polyhistor-lite can further reduce trainable parameters and achieve higher accuracy than all other methods using a similar amount of trainable parameters (e.g., BitFit, PHM layer, Compacter, LoRA, and Low-rank Adapter). This trend is aligned with what we found in the original experiments when using Swin Transformer. With these new experiments, we show that our method generalizes to different backbones.
>
> |   |                  ▼ Results of **PVT**                             |              |                |              |                 |               |
> |:----------------------------:|:--------------------------------------------:|:------------:|:--------------:|:------------:|:---------------:|:-------------:|
> |            **Method**            | **Trainable Parameters (Encoder/All; Millions)** | **Seg.↑ (mIoU)** | **H.Seg.↑ (mIoU)** | **Sal.↑ (mIoU)** | **Normals↓ (mErr)** | **Avg. Improve.** |
> | Single-task full fine-tuning |                  0.00/97.99                  |     68.81    |      61.27     |     62.67    |      17.55      |     0.00%     |
> |     Fine-tuning Decoders     |                   0.00/2.11                  |     64.86    |      51.18     |     61.54    |      19.55      |     -8.85%    |
> |            Bitfit            |                   0.22/2.34                  |     71.41    |      55.71     |     64.08    |      18.69      |     -2.38%    |
> |            Adapter           |                   0.79/2.90                  |     71.94    |      56.38     |     64.16    |      18.75      |     -1.97%    |
> |             LoRA             |                   0.30/2.41                  |     71.89    |      56.90     |     64.27    |      18.48      |     -1.35%    |
> |       Low-Rank adapter       |                   0.25/2.36                  |     70.72    |      55.34     |     63.39    |      18.70      |     -3.08%    |
> |           PHM layer          |                   0.42/2.53                  |     70.81    |      55.02     |     63.51    |      18.75      |     -3.20%    |
> |          Compacter++         |                   0.09/2.20                  |     70.29    |      54.80     |     63.16    |      18.82      |     -3.71%    |
> |          Hyperformer         |                  14.03/16.14                 |     70.81    |      57.76     |     65.49    |      17.75      |     0.14%     |
> |        **Polyhistor**        |                **5.21/7.32**               |    **71.00**    |    **57.52**   |   **65.83**  |    **17.83**    |   **0.13%**   |
> |      **Polyhistor-Lite**     |                 **0.29/2.40**                |   **70.93**  |    **56.71**   |   **65.00**  |    **17.95**    |   **-0.73%**  |
>
> ---
>
> ***Q2: In Figure 2(b), the Transformer in the lower part has no direct connection with the upper part and is meaningless.***
>
> **A:** Thanks for pointing this out, and we modified this figure for better understanding. We intend to show that channel sizes and adapter weight sizes are different in different blocks of the hierarchical vision transformers.
>
> ---
>
> **Reference:**
>
> [1] “Pyramid Vision Transformer: A Versatile Backbone for Dense Prediction without Convolutions”, Wang et al., ICCV’ 2021

---

> > ### Comment · Reviewer_eYWB · 2022-08-06
> > **Reviewer Response**
> >
> > Thank the authors for the response.
> > I have no further questions. After reading the rebuttal and other reviewers' comments, I would like to keep my score at 7.

---

### Official Review · Reviewer_Gyt3 · 2022-07-11

**Rating:** 5
**Confidence:** 4
**Soundness:** 3 good
**Presentation:** 3 good
**Contribution:** 3 good

**Summary:**

The manuscript provides an extensive multi-task parameter-efficient benchmark and examines existing parameter-efficient fine-tuning NLP methods for vision tasks. The main contribution is that this work is the first to address parameter-efficient multi-task adaptation for vision tasks, and developed a unified framework to benchmark several parameter-efficient fine-tuning NLP methods on dense vision tasks. The paper is mostly written in a good manner, and the idea is straight yet effective.

**Questions:**

Major:
1.	The framework design of proposed Polyhistor is not explained clearly. Illustration on main steps of parameters choosing as well as the formulations are encouraged, and the usage of schematic diagram makes the idea easier to follow and understand.
2.	The difference between Visual Prompt Tuning and this work could be compared more concisely.
3.	The novelty of this work mainly focused on the improving HyperNetwork and proposed scalable kernel, thus introduce more about these ideas would be of help.

Minor:
1.	A small grammar mistake: in line 82, it should be ‘a unified framework’, not ‘an’.


**Limitations:**

The true novelty of the work should be further justified.

**Strengths And Weaknesses:**

The paper seems to combine several existing methods, and extend their applicable scene, making the contribution of this work less of a strength. However, they did several modifications to the existing methodologies, which should be detailed and illustrated.

---

> ### Author Response · Authors · 2022-08-02
> **We modified the framework figure, added in-depth discussions on VPT, and added more descriptions on our method**
>
> ***Q1: The framework design of the proposed Polyhistor is not explained clearly. The usage of schematic diagram makes the idea easier to follow and understand.***
>
> **A:** Thanks for the suggestion. We will make some changes in the revised paper to help make it easier to understand.
> - We made the description of Visual prompt tuning more concisely in the related works.
> - We modified the figure of our main framework (Figure 2b) and made the framework clear.
> - We modified the caption of Figure 2 to clarify the framework design of Polyhistor and Polyhistor-Lite.
>
> ---
>
> ***Q2: The difference between Visual Prompt Tuning and this work could be compared more concisely.***
>
> **A:** Thanks for the suggestion. We made the comparison between Visual Prompt Tuning (VPT) and our work more concisely in the main paper (line 125-126).  We also provided the empirical comparison to VPT in all our experimental cases. In addition, we would be happy to provide a more in-depth comparison to VPT in the following paragraphs, and we put these additional discussions in the appendix (Section 1.7 of the revised appendix).
>
>
> - **[Different Problem Settings]** VPT focuses on single-task parameter-efficient adaptation, while our proposed method focuses on multi-task parameter-efficient adaptation. Our goal is to perform a parameter-efficient adaptation for multiple tasks and share the beneficial information across multiple vision tasks.
> - **[Different types of parameter-efficient methods]** VPT adds learnable parameters along with the visual embeddings, while our proposed method utilizes a shared hyper-network to produce the adapter weights for different tasks. Also, the insertion locations of learnable parameters are different (VPT: input space, Ours: parallel to fully-connected layers).
>
> ---
>
> ***Q3: Introducing more about HyperNetwork and the proposed Scalable Kernel could help.***
>
>
> **A:** Thanks for the suggestion. We added more descriptions of Hypernetwork and Scalable Kernel.
> - **[HyperNetwork]** To learn the jointly beneficial information across different visual tasks, we introduce a pair of hyper-networks, which are learnable individual modules, to produce the weights of the adapters inserted in the dense prediction model. Different from the prior work [1], we decompose the adapter weight into two low-rank matrices and thus significantly reduce the parameters used in the hypernetworks (as shown in Section 4.1).
>
> - **[Scaling Kernels]** Scaling Kernels are proposed to address the quadratically growing parameters issue for hierarchical vision transformers, and these layer-wise Scaling Kernels are then combined with the Template Kernels (produced by the hypernetworks) by using Kronecker Produce. In this way, we can efficiently scale up the Template Kernels and fit them into transformer layers with different scales (as shown in Section 4.2).
>
> ---
>
> ***Q4: Typos.***
>
> **A:** Thanks for pointing this out. We have corrected this accordingly.

---

### Official Review · Reviewer_Qanq · 2022-07-11

**Rating:** 7
**Confidence:** 4
**Soundness:** 3 good
**Presentation:** 3 good
**Contribution:** 3 good

**Summary:**

This paper proposes Polyhistor and Polyhistor-Lite, consisting of Decomposed HyperNetworks and Layer-wise Scaling Kernels, to share information across different tasks with a few trainable parameters and address parameter-efficient multi-task adaptation for vision tasks. The authors construct a unified framework with the same implementation details and provide a comprehensive and fair comparison between existing parameter-efficient adaptation works in NLP on multi-tasking dense vision problems. Compared with the state-of-the-art multi-tasking parameter-efficient adaptation method, the method achieves competitive performance improvement with ∼ 90% reduction in the trainable parameters .

**Questions:**

1. In Table 1 in Appendix, why the performance of Polyhistor-Lite (ρ = 32) in semantic segmentation is slightly better than Polyhistor-Lite (ρ = 1). It would be better to show more results of different down-projection ratio of adapters.
2. Can this method be applied to other backbones?

**Limitations:**

The method is novel and solve multiple tasks with limited tunable parameters. However, there are several hyper-parameters needed to be tuned in this method. Besides, The method only focuses on dense vision tasks. It would be better to include more common vision tasks like object detection.

**Strengths And Weaknesses:**

Strengths:
1. This paper conducts a thorough study on how the existing successful parameter efficient methods on NLP tasks perform on vision tasks, i.e., semantic segmentation, human part segmentation, saliency detection, and surface normals estimation.
2. The authors design a novel parameter-efficient method for adaptation to dense vision tasks. Specifically, the hyper-networks take input task embeddings and layer embeddings to produce low-rank matrices and further obtain the adapter weights.
3. Experimental results show that Polyhistor-Lite can achieve a competitive performance gain compared with the state-of-the-art method and only use a very limited amount of tunable parameters.

Weaknesses:
1. There are many hyper-parameters in this method, e.g., task embedding size, dimension of ranks, and the down-projection ratio of adapters. Searching for suitable hyper-parameters costs a lot of resources.
2. The method is implemented based on SwinTransformer backbone. It would be better to conduct experiments on more other backbones.

---

> ### Author Response · Authors · 2022-08-02
> **We added one analysis on adapter down-projection ratios and one experiment for another backbone.**
>
> ***Q1: Results of different down-project ratios of adapters?***
>
> **A:** We vary the down-projection ratios (ρ)  of the adapters and report the results in the Table. We find that the semantic segmentation reaches the near-optimal performance when the small adapters are used (ρ = 32). However, for other dense prediction tasks, there exist obvious gaps when the smaller adapters are used, and averaged relative improvement shrinks when the adapter sizes are smaller.
> This suggests that the required network capacity for semantic segmentation is sufficient when small adapters are used, while other dense prediction tasks require more trainable parameters.
>
> Such a trend potentially comes from the usage of a backbone pretrained on image classification tasks with overlapping object categories (ImageNet). Such a backbone is expected to contain similar semantic information required by semantic segmentation, so that using a limited amount of trainable parameters can achieve near-optimal results.
>
> | Down-Proj.  Ratio |   Methods   | Trainable Parameters (Encoder/ All; Millions) | Seg.↑ (mIoU) | H.Seg.↑ (mIoU) | Sal.↑ (mIoU) | Normals↓ (mErr) | Avg.  Improve. |
> |:-----------------:|:----:|:---------------------------------------------:|:------------:|:--------------:|:------------:|:---------------:|:--------------:|
> |         1         | Ours |                  1.291/1.3353                 |     73.7     |      63.32     |     66.5     |      16.93      |      6.38%     |
> |         2         | Ours |                 0.6213/3.6656                 |     73.69    |      63.04     |     66.56    |      17.301     |      5.80%     |
> |         4         | Ours |                 0.3862/3.4305                 |     73.57    |      62.04     |     65.84    |       17.7      |      4.55%     |
> |         8         | Ours |                  0.2937/3.338                 |     73.92    |      62.15     |     65.37    |       17.7      |      4.53%     |
> |         32        | Ours |                 0.2352/3.2795                 |     73.8     |      61.32     |     64.64    |      17.92      |      3.57%     |

---

> > ### Author Response · Authors · 2022-08-02
> > **Cont'd**
> >
> > ***Q2: Can the proposed method be applied to other backbones?***
> >
> > **A:** Yes, our method can be applied to other backbones. In addition to applying it to the Swin Transformer in the paper, for the rebuttal we further apply our method and other baseline methods to the **Pyramid Vision Transformer** [1] as shown in the Table. The conclusions are consistent with our original experiments. We find our Polyhistor can achieve comparable results to Hyperformer while using much fewer trainable parameters. Polyhistor-lite can further reduce trainable parameters and achieve higher accuracy than all other methods using a similar amount of trainable parameters (e.g., BitFit, PHM layer, Compacter, LoRA, and Low-rank Adapter). This trend is aligned with what we found in the original experiments when using Swin Transformer. With these new experiments, we show that our method generalizes to different backbones.
> >
> > |  |                   ▼ Results of **PVT**                            |              |                |              |                 |               |
> > |:----------------------------:|:--------------------------------------------:|:------------:|:--------------:|:------------:|:---------------:|:-------------:|
> > |            **Method**            | **Trainable Parameters (Encoder/All; Millions)** | **Seg.↑ (mIoU)** | **H.Seg.↑ (mIoU)** | **Sal.↑ (mIoU)** | **Normals↓ (mErr)** | **Avg. Improve.** |
> > | Single-task full fine-tuning |                  0.00/97.99                  |     68.81    |      61.27     |     62.67    |      17.55      |     0.00%     |
> > |     Fine-tuning Decoders     |                   0.00/2.11                  |     64.86    |      51.18     |     61.54    |      19.55      |     -8.85%    |
> > |            Bitfit            |                   0.22/2.34                  |     71.41    |      55.71     |     64.08    |      18.69      |     -2.38%    |
> > |            Adapter           |                   0.79/2.90                  |     71.94    |      56.38     |     64.16    |      18.75      |     -1.97%    |
> > |             LoRA             |                   0.30/2.41                  |     71.89    |      56.90     |     64.27    |      18.48      |     -1.35%    |
> > |       Low-Rank adapter       |                   0.25/2.36                  |     70.72    |      55.34     |     63.39    |      18.70      |     -3.08%    |
> > |           PHM layer          |                   0.42/2.53                  |     70.81    |      55.02     |     63.51    |      18.75      |     -3.20%    |
> > |          Compacter++         |                   0.09/2.20                  |     70.29    |      54.80     |     63.16    |      18.82      |     -3.71%    |
> > |          Hyperformer         |                  14.03/16.14                 |     70.81    |      57.76     |     65.49    |      17.75      |     0.14%     |
> > |        **Polyhistor**        |                **5.21/7.32**               |    **71.00**    |    **57.52**   |   **65.83**  |    **17.83**    |   **0.13%**   |
> > |      **Polyhistor-Lite**     |                 **0.29/2.40**                |   **70.93**  |    **56.71**   |   **65.00**  |    **17.95**    |   **-0.73%**  |
> >
> >
> >
> >
> > **Reference:**
> >
> > [1] “Pyramid Vision Transformer: A Versatile Backbone for Dense Prediction without Convolutions”, Wang et al., ICCV’ 2021

---

### Author Response · Authors · 2022-08-02
**Summary of rebuttal**

We thank all reviewers for providing constructive thoughtful feedback!

We are deeply encouraged by the reviewers’ positive comments such as *“conducts a thorough study”* (R #Qanq, R #eYWB), *“novel method”* (R #Qanq), *“reasonable method”* (R #eYWB, R #zAUZ), *“achieves a competitive performance gain”* (R #Qanq), *“interesting to explore the parameter-efficient methods for dense vision tasks”* (R #eYWB), *“presentation is clear”* (R #eYWB, R #zAUZ),  and *“benchmark is helpful to the community”* (R #zAUZ).

---

We appreciate the above positive comments and would like to provide more experiments and analyses to further improve our paper. We summarize the additional experiments/analyses we made in the rebuttal per the suggestions from the reviewers.

- **[Experiment added]** We show our proposed Polyhistor and Polyhistor-Lite can be applied to other backbone architectures (e.g., Pyramid Vision Transformer [1]), and our proposed methods can achieve comparable results to the SoTA method (i.e., Hyperformer) by using significantly fewer trainable parameters.
- **[Experiment added]**  We show our proposed Polyhistor and Polyhistor-Lite can be applied to self-supervised models (e.g.,  MoBY [2]; self-supervised SwinTransformer), and our proposed methods can achieve competitive or even better results against the other methods with fewer trainable parameters.
- **[Experiment added]** We examine our proposed method under sequential multi-tasking learning.
- **[Analysis added]**  We vary different down-projection ratios of adapters and report their results on multiple vision tasks.

---

In addition, we revised our paper as listed in the following points, and we colorize them blue in the revised version.

- We modified Figure 2 and its caption for a better understanding of our framework.
- We made the comparison to VPT more concisely and put an in-depth discussion in the appendix.
- We added the above new experiments and their discussion in the appendix.

Thank you again for your time and effort in reviewing our paper!


---

Reference:

[1] “Pyramid Vision Transformer: A Versatile Backbone for Dense Prediction without Convolutions”, Wang et al., ICCV’ 2021

[2] “Self-Supervised Learning with Swin Transformers”,  Xie et al., arXiv 2021

---

### Meta-Review · Area_Chair_6WaD · 2022-08-26

**Recommendation:** Accept
**Confidence:** Certain

**Metareview:**

The proposed Polyhistor and Polyhistor-Lite for parameter-efficient multi-task adaptation achieves competitive performance gains on dense vision datasets. All reviewers give consistent positive scores. The requested experiments for more backbones, self-supervised backbones and analyses have been accordingly added during the discussion phase. Reviewer Gyt3 is concerned about the unclear explanation of the framework, and why HyperNetwork and scalable kernels could help. The authors addressed the issues and modified the paper. The meta-reviewers thus recommend to accept this paper, and encourage the authors to add all new experiments and make the presentation more clear in the camera ready.

**Award:**

No

---

### Decision · Program_Chairs · 2022-09-14

Accept